Vertical jump performance in recreational runners with visual impairment: a cross-sectional study

da Silva Edson Soares 1 2
De Araújo Pedro Henrique 3
Mindrescu Veronica 4
Liedtke Fabio 2
http://orcid.org/0000-0003-1742-5016 Peyré-Tartaruga Leonardo Alexandre 2 5
Fischer Gabriela 3 gabriela.fischerrs@gmail.com
1 Inter-University Laboratory of Human Movement Biology, Université Jean Monnet , Saint-Etienne , France
2 LaBiodin Biodynamics Laboratory, Escola de Educação Física, Fisioterapia e Dança, Universidade Federal do Rio Grande do Sul , Porto Alegre, Rio Grande do Sul , Brazil
3 Laboratório do Biomecânica, Centro de Desportos (CDS), Universidade Federal de Santa Catarina , Florianópolis, Santa Catarina , Brazil
4 Motor Performance Department, Transilvania University of Brasov , Brașov , Romania
5 Department of Public Health, Experimental Medicine and Forensic Sciences, University of Pavia , Pavia , Italy
Redondo Beatriz
Electronic publication date: 2025 Mar 18
Publication date: 2025
Volume: 13
Electronic Location ID: e19059
Received 2023 Dec 14; Accepted 2025 Feb 5
Copyright: © 2025 da Silva et al.
Copyright year: 2025
Copyright holder: da Silva et al.
License: This is an open access article distributed under the terms of the Creative Commons Attribution License, which permits unrestricted use, distribution, reproduction and adaptation in any medium and for any purpose provided that it is properly attributed. For attribution, the original author(s), title, publication source (PeerJ) and either DOI or URL of the article must be cited.
License URL: https://creativecommons.org/licenses/by/4.0/

Keywords: Blind, Kinematic, Angle, Lower limb, Power

Funding: The authors received no funding for this work.

==============================
Background

Runners with visual impairment (VI) may show changes in jump technique due to momentary loss of spatial reference specifically related to the base of support on the ground and rapid postural adjustment. The vertical jump performance and the analysis of the technique can provide information about the neuromuscular characteristics of the runners with VI, motor control and training strategies.

Objective

Thus, the purpose of this study was to analyze the relationship between vertical jump performance and lower limb joint angles in recreational runners with VI.

Methods

Eight recreational runners (age 33.9 ± 12.7 years and body mass 64.1 ± 13.6 kg) with VI (visual classification: 4 T1, 2 T2 and 2 T3) performed five consecutive squat jumps (SJ) and five consecutive countermovement jumps (CMJ) with 5 min of rest between them. The vertical jumps were recorded by one camera and the jump height and power were evaluated using accelerometer Myotest.

Results

No difference was found between the SJ (16.8 ± 4.9 cm) and CMJ performances (13.6 ± 5.8 cm) (p = 0.056). Pearson’s correlation test identified a strong and negative correlation between SJ height and internal knee angle (r = −0.712; p = 0.047), while no correlation between CMJ height and internal knee angle was found (r = −0.226; p = 0.591).

Conclusion

We concluded that recreational runners with VI reach higher jump heights in a deeper squat position during SJ.

Introduction

Jumping is a fundamental movement pattern essential in numerous sports. In discipline such as basketball, soccer and volleyball, vertical jump performance plays a critical role in determining scores and possibly matches (Rodríguez-Rosell et al., 2017). Moreover, vertical jump assessments serve as powerful tool to evaluate athletic capabilities in both Olympic and Paralympic contexts (Freitas et al., 2022).

Key parameters such as vertical jump height and power output are have been shown to correlated with performance across various sports (Berriel et al., 2021; Kons et al., 2017; Loturco et al., 2015; Pereira et al., 2016). The ability to accurately assess these parameters provides valuable data for training and workload management, not only for able-bodied (Loturco et al., 2022; Marco-Contreras et al., 2021) but also for visual impaired (VI) athletes (Loturco et al., 2015). While force platform or 3D motion capture system are considered the gold standard to obtaining such measurements with precision, a variety of more accessible, field-based alternatives have emerged. Devices such as accelerometers, contact mats, photocells and smartphone applications have gained popularity among researchers and coaches due to their validity and reliability in practical (Balsalobre-Fernández, Glaister & Lockey, 2015; Watkins et al., 2020).

In the context of countermovement jump (CMJ), there appears to be an optimal squat position for maximizing performance (Mandic, Jakovljevic & Jaric, 2015). Some studies have demonstrated that a greater descent during the eccentric phase leads to enhanced performance outcomes (Lees, Vanrenterghem & Clercq, 2004; Pérez-Castilla et al., 2021; Sánchez-Sixto, Harrison & Floría, 2018). Interestingly, a similar relationship between deeper squat depth and performance has been observed in the squat jump (SJ). However, self-selected squat positions have shown better reliability in SJ performance (La Torre et al., 2010; Mitchell et al., 2017; Petronijevic et al., 2018). These findings emphasize the distinct neuromuscular demands represented by CMJ and SJ, which assess different aspect of an athlete’s physical capabilities (Van Hooren & Zolotarjova, 2017).

Currently, an estimated 43 million worldwide live with visual impairment (GBD 2019 Blindness and Vision Impairment Collaborators, 2021). Individuals with VI face significant challenges in performing activities, which often result in reduced activity levels (Cai et al., 2021) and decrease life expectancy (McCarty, Nanjan & Taylor, 2001). Participation in sport has been shown to be benefit VI individuals, improving their quality of life, fostering social inclusion (Ilhan, Idil & Ilhan, 2021), and enhancing self-selected walking speed (Silva et al., 2018). In Paralympic competition, athlete is classified into sports classes based on the severity of their visual impairment, ensuring fair competition among individuals with similar functional limitations. In para-athletics, VI athletes are assigned to classes T11, T12, and T13, as defined by the International Blind Sports Federation (2018).

It has been well-documented that VI athletes exhibit deficits in power, speed and strength compared to their sighted counterparts, both in jumping and sprinting task (Freitas et al., 2022; Pereira et al., 2016). Vertical jump height, for instance, is determined by relative net propulsive impulse generated during the jump (Kirby et al., 2011). Furthermore, VI athletes may adopt altered squat positions in order to compensate for their visual impairment during the execution of CMJ and SJ. However, the impact of visual impairment on lower limb kinematics during vertical jump remains largely unexplored. While, somatosensorial feedback may compensate for the loss of visual information (Gipsman, 1981), the specific mechanical determinants and performance characteristics of VI athletes during jumping task remain poorly understood (Haibach, Wagner & Lieberman, 2014). Thus, the aim of this study is to examine the relationship between vertical jump performance and lower limb joint angles in recreational runners with VI. We hypothesize that, due to the great complexity of the CMJ movement pattern (Pereira et al., 2016) and the enhanced self-perception of safety associated with the SJ (Kons et al., 2019), recreational runners with VI will adopt a deeper squat position when performing SJ compared to the CMJ.

Materials and Methods

Subjects

Eight recreational runners (three male and five female) with visual impairment (visual classification: 4 T1, 2 T2 and 2 T3) participated in this study. The participants had a mean age 33.9 ± 12.7 years, a body mass 64.1 ± 13.6 kg, and an average height 1.69 ± 0.07 m. Visual classifications followed the criteria set by the International Blind Sports Federation (2018), where T1 the visual acuity is less than LogMAR 2.60; T2 includes acuity between LogMAR 1.50 and 2.60 or a visual field restricted to less than 10 degrees; and T3 includes acuity between LogMAR 1 and 1.40 or a visual field constricted to less than 40 degrees. Participants had a weekly running training load of 4.9 ± 3.9 h. Eligibility criteria required participants to be free from chronic joint pain and musculoskeletal or bone injuries within the 6-month preceding the study. Participants were recruited through non-probabilistic sampling. The study received ethical approval from the Universidade Federal do Rio Grande do Sul (CAE: 69344117.2.0000.5347), and informed consent was obtained from all participants, with the consent form read aloud by one of the researchers.

Study design

This is a cross-sectional study followed the STROBE checklist guidelines (von Elm et al., 2008). Familiarization and experimental session were conducted at the same day. Participants completed a 5-min run at a self-selected running speed on a 400 m track. Familiarization involved 10 min of guided jump practice, led by an experienced instructor, at a non-slippery and quiet location near their training area. The instructor provided verbal cues and guided participants’ body positions, including trunk inclination, internal knee angle, landing phase, and instruction to: “push the floor”. Following this, participants performed five consecutive squat jumps and five consecutive countermovement jumps with 5-min resting between SJ and CMJ trials.

Vertical jump and lower limb joint angles

After a 5-min running warm-up, participants underwent a 10-min structured familiarization with vertical jump. They then, completed five consecutive SJ and five consecutive CMJ with 5-min rest intervals (Casartelli, Muller & Maffiuletti, 2010). The jumps were recorded using a camera (Nikon, Coolpix L120) with a sampling frequency of 60 Hz, positioned 2 m from the sagittal plane.

Performance data for SJ and CMJ were collected using the Myotest device (5.4 × 10.2 × 11.1 cm and weight: 58 g) with sampling frequency of 500 Hz. The Myotest device has demonstrated high reliability with an intraclass correlation coefficient of 0.98 when compared with Optojump for jump height (Casartelli, Muller & Maffiuletti, 2010) and 0.88 when compared with force platform for CMJ flight time (Castagna et al., 2013).

For the analysis of lower limb angles, five markers were placed on the right side of body at the shoulder, greater trochanter, knee, lateral malleolus, and 5th metatarsal (Fig. 1). Joint angles of the hip (shoulder, greater trochanter, and knee), knee (greater trochanter, knee and lateral malleolus) and ankle (knee, lateral malleolus, and 5th metatarsal) during contact were calculated. For SJ, angles during the static preparation phase were analyzed, while for CMJ, angles were measured at the transition between the eccentric and concentric phases. Full knee extension was set at 180°, and decreasing angle indicating increasing flexion. Joint parameters were digitalized and calculated using Kinovea® v.0.8.15 software (Fernández-González et al., 2020).

Figure 1 Lower limb angles and vertical jump performance assessment.

Hip, knee, and ankle angles are demonstrated from top to bottom.

The CMJ/SJ ratio, representing the efficacy of the stretch-shortening cycle during vertical jump, was calculated by dividing CMJ height by the SJ height (McGuigan et al., 2006).

Statistical analysis

Descriptive statistic, including mean, standard deviation, and 95% confidence interval, were calculated. Normality was assessed using Shapiro-Wilk test. Paired samples t-test were used to compare SJ and CMJ performance. Effect size (ES, Glass’s delta) was calculated to quantify the magnitude of the difference between the two vertical jump types, with effect classifications as follows: null (<0.1), very small (0.1 to 0.19), small (0.2 to 0.49), medium (0.5 to 0.79), large (0.8 to 1.19), very large (1.2 to 1.9), and huge (over 2.0). Pearson product-moment correlation was used to analyze the relationship between vertical jump performance and lower limb joint angle. Correlation coefficients were classified as null (r = 0), weak (0 to 0.3), moderate (0.3 and 0.6), strong (0.6 and 0.9), very strong (0.9 and 1), and perfect (1) (Hopkins, 2000). All descriptive and inferential analyses were conducted using JASP software version 0.16 (JASP Team, 2024) with statistical significance set at α = 0.05. Post-hoc power analysis was done to estimate type 2 error probability (1 – β) using GPower software version 3.1 (Kiel, Germany).

Results

Eight recreational runners with visual impairment participated of this study. Comparisons between SJ and CMJ performance variables and lower limb angles are presented in Table 1.

Table 1 Vertical jump performance and lower limb angles.

					(95%) CI	
Variables	SJ	CMJ	p-value	ES	Lower	Upper	
Jump height (cm)	16.8 ± 4.9	13.6 ± 5.8	0.056	−0.65	−0.102	6.500	
Jump power (W/kg)	27.2 ± 5.7	23.5 ± 7.6	0.128	−0.65	−1.374	8.774	
Hip angle (°)	86.8 ± 12.5	80.9 ± 9.4	0.164	−0.47	−3.030	14.680	
Knee angle (°)	88.1 ± 6.3	84.9 ± 7.6	0.099	−0.51	−0.777	7.177	
Ankle angle (°)	87.1 ± 3.8	87.9 ± 4.5	0.365	0.21	−2.838	1.188	
Note:

Confidence interval (CI); Effect size (ES) represented by Delta Glass.

The CMJ/SJ ratio was 0.8 ± 0.3. No significant differences were found between SJ and CMJ height or power (Table 1). Additionally, hip, knee, and ankle joint angles showed no significant changes between SJ and CMJ (Table 1).

As expected, a strong positive correlation was observed between SJ height and SJ power (r = 0.891; p = 0.003), and very strong correlation was found between CMJ height and CMJ power (r = 0.969; p < 0.001). A strong negative correlation was found between SJ height and knee angle (r = −0.712; p = 0.047), while no significant correlation was found between CMJ height and knee angle (r = −0.226; p = 0.591) as shown in Fig. 2. No significant correlations were observed between SJ or CMJ height and hip and ankle angles.

Figure 2 Bivariate correlations.

Correlation between internal knee angle and SJ height (A) and CMJ height (B). Gray areas show the intervals of confidence at 95%.

Figure 3 compares the SJ and CMJ heights from our study with those of Paralympic athletes at various performance levels.

Figure 3 Comparison between SJ and CMJ in different running level performance.

Eight recreational runners with VI with visual classification T11, T12 and T13 from present study; 10 Paralympic sprinters with visual impairment (PSVI) with visual classification T11 vs 10 respective guides (Pereira et al., 2016); 15 PSVI with visual classification T11 and T12 vs 12 Olympic sprinters (Freitas et al., 2022). * represents significant differences between groups in the cited studies. All groups are represented by shades of gray.

Post-hoc power analysis revealed that the measurements of jump height (power = 0.51) and mechanical power (power = 0.32) were underpowered, indicating a higher probability of type 2 error in detecting differences where none were found.

Discussion

The aim of this study was to investigate the relationship between vertical jump performance and lower limb joint angles in recreational runners with VI. We hypothesized that a more flexed knee during SJ would correlate with better performance due the motion pattern complexity of CMJ jump (Pereira et al., 2016) and greater perceived safety in SJ execution (Kons et al., 2019). Our results confirmed a strong negative correlation between SJ height and knee angle, indicating that recreational runners with VI achieve higher jump heights when adopting a deeper squat position, in line with our hypothesis.

The relationship between squat depth and vertical jump performance remains unclear, as different squat positions can alter muscle-tendon unit lengths, joint moment arms, and thus affect force production (Bobbert, Casius & Kistemaker, 2013). Most studies on sighted individuals show that deeper squat position tend to enhance vertical jump performance (Kirby et al., 2011; La Torre et al., 2010; McBride et al., 2010; Moran & Wallace, 2007; Salles, Baltzopoulos & Rittweger, 2011; Gheller et al., 2015). However, Mitchell et al. (2017) found optimal jump heights at internal knee angles close to 90°.

The mechanics of vertical jump performance depend on several factors, including rate of force development, peak power, vertical impulse, joint kinetics and training background as well as muscle mechanical proprieties (Earp et al., 2010; Kobal et al., 2017; McErlain-Naylor, King & Pain, 2014; McLellan, Lovell & Gass, 2011; Ugrinowitsch et al., 2007; Vanezis & Lees, 2005). While deeper squats can improve mechanical parameters, such as net impulse and peak power, individuals may still choose a self-selected squat position that may or may not maximize jump height (Gheller et al., 2015; Kirby et al., 2011). Thus, any loss or improvement of vertical jump performance may lie in these parameters. Indeed, when individuals adopt a deeper squat position, some mechanical parameters are enhanced, thus resulting in a greater jump height. Gheller et al. (2015) found that in a deeper squat position (CMJ < 90° and CMJ at self-selected position), individuals apply greater relative net impulse and jump height and greater peak power and maximal force at CMJ > 90° and SJ perform parallel CMJ changes. These results are in line with Kirby et al. (2011) who demonstrated relative net vertical impulse during propulsive phase is a strong predictor of jump height for SJ (r = 0.93) and CMJ (r = 0.92). Even with enhancement of some mechanical parameters, it will not necessarily lead to a greater jump height. Interestingly, some studies show that the vertical jump performance is increased or not different when the subjects choose a self-selected squat position than the optimum (Gheller et al., 2015; Kirby et al., 2011; Mandic, Jakovljevic & Jaric, 2015; Mitchell et al., 2017).

Despite extensive research on sighted individuals, limited studies have explored vertical jump performance in VI athletes, who generally exhibit lower jump height and power outputs compared to their sighted counterparts (Pereira et al., 2016; Freitas et al., 2022). For example, Paralympic sprinters classified as T11 and T12 show significantly lower SJ and CMJ height (20% and 19% difference, respectively) as well as lower power output during half and squat jumps (32% and 20% difference, respectively) than Olympic sprinters (Pereira et al., 2016). All these jump heights in Paralympic sprinters are more than two times higher than recreational runners with VI in the present study (see Fig. 3). Possibly, part of these differences may rely substantially on individuals’ training/activity background, which is known to influence muscle architecture and, thereby, explosiveness and jump performance (Laroche et al., 2007; Kobal et al., 2017; Ugrinowitsch et al., 2007). In line, an interesting study showed that goalball players with VI are superior in a set of physical fitness tests, that included CMJ, compared to non-goalball players with VI (Çolak et al., 2004). Therefore, there are differences between sighted and VI individuals, for both power and functional parameters, and it seems that these differences tend to be lesser in a greater activity or training level.

Interestingly, task performance appears to change when spatial orientation is required in individuals with VI. Pereira et al. (2016) compared vertical and horizontal jump performance between paralympic sprinters (T11) and their guides. They found that, when jump orientation was vertically, the difference magnitudes were lower than when the task was to perform horizontally. Also, this “directional preference” can be observed in another study carried out by Ray et al. (2007), which observed significant differences between VI and sighted individuals in horizontally oriented tasks (walk and forward lunge), but not in vertically oriented (sit to stand test). Interestingly, these findings are not confined only to spatial orientation but also to higher demanding and higher velocity tasks. Loturco et al. (2017) observed a detrimental effect of VI in mean propulsive power for jump squat, bench press, and standing barbell row but not for maximal isometric strength between Olympic and Paralympic athletes. To sum up, a study carried by Kons et al. (2019) shows a larger difference between sighted and VI judo athletes in CMJ than in SJ. From a mechanistic point of view, CMJ is a higher velocity task and demands a larger coordination than SJ due its motor complexity. Therefore, greater sensory input may be needed to anticipate the countermovement transition and provide a greater explosiveness.

Therefore, in a task where the center of mass is shifted out of its base of support and/or rapid adjustments have to be done, individuals with VI present impaired task performance, possibly due to a more cautious strategy (Kons et al., 2019). This strategy must be attributable to a better self-perception of safety (only concentric phase for SJ vs. eccentric/concentric phase for CMJ), as a result of the impairments in the visual system since it has a big role in dynamic postural control. For this reason, a non-significant difference and a CMJ/SJ ratio lower than 1 were observed, which is in line with the findings of the study carried out by Kons et al. (2019). Besides the reduced visual control mechanisms impairing jumps with faster and more complex gestures as in the CMJ compared to the SJ (Iguchi, Nozu & Sakuma, 2022), it is worth noting that in people with VI trained in jumping, the difference is larger, indicating that the impairment-inducing role is not reduced with jump training (Killebrew et al., 2013).

To our knowledge, this is the first study relating the vertical jump performance to joint angles in VI recreational runners. The limitation of this study is the low sample size and heterogeneity of visual classification. It was included four subjects T1, two subjects T2 and two subjects T3 who are recreational runners. Thus, the underpowered results. Additionally, it is important to report that men and women were used and, then, it can add additional variability in lower limb power. Considering that power training is an appropriate method for attaining higher metabolic economy and performance for distance runners (Ramirez-Campillo et al., 2021), and that power training may be easily applied in real training setups, what still remains to be established is whether power training might be applied in VI recreational runners. Our findings indicate that squat jump can be a good option for VI recreational runners. Further studies evaluating different phases of vertical jump (i.e., relative net vertical impulse, peak of force, power, and velocity), static and dynamic balance during vertical jump, as well as including the sighted control group, are needed to overcome confounding factors. These findings have implications for the organization of exercises and training periodization for VI recreational runners.

Conclusions

We concluded recreational runners with VI reach higher jump heights in a deeper squat position for SJ, and the performance of the SJ and CMJ are similar.

Supplemental Information

Supplemental Information 1 Raw data.

Supplemental Information 2 STROBE checklist.

We thank to Associação de Cegos do Rio Grande do Sul–ACERGS for the subject’s participation. Additionally, we would like to express our gratitude to dablio.rodrigues who created Fig. 1 of this study as part of an initiative to promote original artwork in scientific articles by black Brazilian artists.

Additional Information and Declarations

Competing Interests

Leonardo Alexandre Peyré-Tartaruga is an Academic Editor for PeerJ.

Author Contributions

Edson Soares da Silva conceived and designed the experiments, performed the experiments, analyzed the data, prepared figures and/or tables, authored or reviewed drafts of the article, and approved the final draft.

Pedro Henrique De Araújo analyzed the data, prepared figures and/or tables, authored or reviewed drafts of the article, and approved the final draft.

Veronica Mindrescu analyzed the data, authored or reviewed drafts of the article, and approved the final draft.

Fabio Liedtke conceived and designed the experiments, performed the experiments, analyzed the data, authored or reviewed drafts of the article, and approved the final draft.

Leonardo Alexandre Peyré-Tartaruga conceived and designed the experiments, analyzed the data, authored or reviewed drafts of the article, and approved the final draft.

Gabriela Fischer conceived and designed the experiments, performed the experiments, analyzed the data, prepared figures and/or tables, authored or reviewed drafts of the article, and approved the final draft.

Human Ethics

The following information was supplied relating to ethical approvals (i.e., approving body and any reference numbers):

The study was approved by the Universidade Federal do Rio Grande do Sul ethics committee.

Data Availability

The following information was supplied regarding data availability:

The raw data are available in the Supplemental File.

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
