# Peer review of "Vertical jump performance in recreational runners with visual impairment: a cross-sectional study"

_PeerJ, doi:10.7717/peerj.19059_

## Round 0.1 · original submission · Major Revisions

· Academic Editor

Major Revisions

Your manuscript has been reviewed by two experts in the field. Revisions are
necessary before the manuscript is suitable for publication.

**Language Note:** The review process has identified that the English language must be improved. PeerJ can provide language editing services - please contact us at [email protected] for pricing (be sure to provide your manuscript number and title). Alternatively, you should make your own arrangements to improve the language quality and provide details in your response letter. – PeerJ Staff

Reviewer 1 ·

Basic reporting

This paper describes the relationship between vertical jump performance and lower limb joint angles in recreational runners with VI. The introduction is clear and a with an adequate literature review. In addition, the authors reflect the objective of the study and the hypotheses.
In the abstract, the authors should indicate the meaning of SJ and CMJ the first time it appears.
The actual classes for VI individuals in para-athletics, if the sport is not specified, are B1, B2 and B3.The authors should refer to the sport, in this case athletics, to indicate the T1, T2 and T3 classes.

Experimental design

I consider that there are some methological limitations that affect the validity and generalizability of the results. Although the method is well described, I consider that the study sample is too small and the results should be compared with a control group of recreational runners without VI. Have the authors used any test to determine the required sample size?
Were the measurements always taken in the same order or randomly?

Validity of the findings

The results shown that visually impaired recreational runners reach higher jump heights in a deeper squat position but the lack of a control group does not allow us to know whether this relationship also exists in the absence of visual impairment.

Figure 3 shows the comparison between SJ and CMJ in different running level performance. Given that the current study includes only recreational runners, it is to be expected that the performance obtained will be lower compared to Paralympic and Olympic athletes.

The contribution to the scientific literature is limited, and there is no clear implication of the findings in the current understanding of the problem addressed.

Additional comments

The authors should indicate the limitations of the study.

Reviewer 2 ·

Basic reporting

The English language should be improved to ensure that an international audience can clearly understand your text. In some sentences, it is necessary to correct the tense, as well as to use the appropriate adverbs. I suggest you have a colleague who is proficient in English and familiar with the subject matter review your manuscript at lines:
77, 90/91, 121, 131, 152/153, 173/174, 178, 210/211, 259/260.

Experimental design

It is pretty unclear if participants performed 5 repetitions of SJ and CMJ, or they were performed series of 5 jump repetitions (lines 123/124)

Validity of the findings

No comment

Additional comments

It is necessary to correct the English language and make the text more understandable for the readers. Methodologically, the test protocol should be better described in order to understand whether subjects jump 5 single jumps, or a series of 5 jumps. Is the device used in the study suitable for subjects with VI and has it already been used for testing?

---

## Round 0.2 · Minor Revisions

· Academic Editor

Minor Revisions

I have now had the opportunity to read your revised manuscript, and your responses to the reviewers' comments. I believe that you have addressed the concerns raised, and I am happy in principle to accept your manuscript for publication in PeerJ. The decision has however been listed as "minor revision" because before scheduling it for publication, I would be grateful if you could address some minor points that are shown.

The authors' explanation of the sample size is insufficient. Even if they calculated it, the sample size is very small, and there is significant heterogeneity in the visual classification of participants. To calculate the sample size, the authors assumed an alpha level of 0.05 and an effect size of 0.61/0.81, which are quite high. Additionally, the statistical power is very low (0.32/0.50). Was the calculation based on a similar study? The authors should include the sample size calculation in the Methods section and acknowledge the low sample size as a limitation in the Limitations section.

---

## Round 0.3 · accepted · Accept

· Academic Editor

Accept

I have now had the opportunity to read your revised manuscript. I believe that you have addressed the concerns raised, and I am happy to accept your manuscript.